# Metabolic Signature of Energy Metabolism Alterations and Excess Nitric Oxide Production in Culture Media Correlate with Low Human Embryo Quality and Unsuccessful Pregnancy

**DOI:** 10.3390/ijms24010890

**Published:** 2023-01-03

**Authors:** Romina Pallisco, Giacomo Lazzarino, Gabriele Bilotta, Francesca Marroni, Renata Mangione, Miriam Wissam Saab, Maria Violetta Brundo, Alessandra Pittalà, Giuseppe Caruso, Elena Capoccia, Giuseppe Lazzarino, Barbara Tavazzi, Pasquale Bilotta, Angela Maria Amorini

**Affiliations:** 1Alma Res Fertility Center, Laboratory of Andrology and Embriology, Via Parenzo 12, 00198 Rome, Italy; 2Departmental Faculty of Medicine and Surgery, UniCamillus—Saint Camillus International University of Health and Medical Sciences, Via di Sant’Alessandro 8, 00131 Rome, Italy; 3Department of Basic Biotechnological Sciences, Intensive and Perioperative Clinics, Catholic University of Rome, Largo F. Vito 1, 00168 Rome, Italy; 4Department of Biomedical and Biotechnological Sciences, Division of Medical Biochemistry, University of Catania, Via S. Sofia 97, 95123 Catania, Italy; 5Department of Biology, Geology and Environmental Sciences, Section of Animal Biology, University of Catania, Via Androne 81, 95124 Catania, Italy; 6Department of Drug and Health Sciences, University of Catania, Viale A. Doria 6, 95125 Catania, Italy; 7LTA-Biotech srl, Viale Don Orione 3D, 95047 Paternò, Italy; 8Alma Res Fertility Center, Obstetrics and Gynecology, Via Parenzo 12, 00198 Rome, Italy

**Keywords:** assisted reproduction techniques, biomarkers, energy metabolites, fertility rates, human embryo, hypoxanthine, infertility, in vitro fertilization, nitrate, nitrite, oxypurines, targeted metabolomics, uric acid

## Abstract

Notwithstanding the great improvement of ART, the overall rate of successful pregnancies from implanted human embryos is definitely low. The current routine embryo quality assessment is performed only through morphological criteria, which has poor predictive capacity since only a minor percentage of those in the highest class give rise to successful pregnancy. Previous studies highlighted the potentiality of the analysis of metabolites in human embryo culture media, useful for the selection of embryos for implantation. In the present study, we analyzed in blind 66 human embryo culture media at 5 days after in vitro fertilization with the aim of quantifying compounds released by cell metabolism that were not present as normal constituents of the human embryo growth media, including purines, pyrimidines, nitrite, and nitrate. Only some purines were detectable (hypoxanthine and uric acid) in the majority of samples, while nitrite and nitrate were always detectable. When matching biochemical results with morphological evaluation, it was found that low grade embryos (n = 12) had significantly higher levels of all the compounds of interest. Moreover, when matching biochemical results according to successful (n = 17) or unsuccessful (n = 25) pregnancy, it was found that human embryos from the latter group released higher concentrations of hypoxanthine, uric acid, nitrite, and nitrate in the culture media. Additionally, those embryos that developed into successful pregnancies were all associated with the birth of healthy newborns. These results, although carried out on a relatively low number of samples, indicate that the analysis of the aforementioned compounds in the culture media of human embryos is a potentially useful tool for the selection of embryos for implantation, possibly leading to an increase in the overall rate of ART.

## 1. Introduction

The rate of couples affected by infertility problems, where infertility is defined as the failure to achieve a clinical pregnancy after at least 12 months of regular unprotected sexual intercourses with the same partner, is constantly increasing, and unfortunately, an increasing number of young couples (within 30 years of age) are affected [1]. The last chance for an infertile couple is to appeal to Advanced Reproductive Techniques (ART) in order to try to overcome fertility problems and obtain a clinical pregnancy ending with healthy offspring. Notwithstanding the improvement of ART, the 2018 report of the Italian Institute of Health (Istituto Superiore di Sanità, ISS), on the activity of the centers practicing medical assisted procreation on the Italian territory, indicated that with an overall number of 97,508 treatment cycles (independently of the ART used), the number of clinical pregnancies was 18,994 (19.5%) and that of healthy newborns was only 14,139, i.e., a success rate of 14.5% [2]. 

One of the relevant parameters that certainly contributes to the low rate of successful clinical pregnancy with ART is the inability to directly characterize human embryos from a biochemical/metabolic point of view with the aim of performing a targeted selection based on objective, quantifiable biological parameters capable of choosing the best embryo for the subsequent transplant. In fact, the many morphological and morphokinetic evaluations of human embryos [3,4,5] do not allow for the measurement of any molecular parameter reflecting metabolism and/or biochemical processes connected to the quality of the embryo. On the other hand, currently there are no analytical techniques capable of determining, noninvasively, biochemical parameters connected to energy metabolism, mitochondrial functions, oxidative stress, or nitric oxide metabolism directly in human embryo cell cultures, rendering it impossible to provide direct evaluation of human embryo metabolism, which is potentially crucial to improving the rate of successful ART. 

The idea of monitoring the human embryo quality through the analysis of metabolites in the embryo culture medium has been applied in the past in numerous studies, initially dedicated mostly to evaluating the consumption of few substrate(s) [6,7,8] and lately, with the introduction of more sensitive methodologies, dedicated to determining a wider number of compounds present in the human embryo culture medium [9,10,11]. In the majority of these recent studies, attention was dedicated to monitoring changes in glucose and amino acids with the aim of indirectly evaluating the metabolic rates of human embryos and their potential connection with successful clinical pregnancy.

Several studies demonstrated that, in the case of an imbalance of energy metabolism, there is a significant increase in the cell/tissue production and release of dephosphorylated ATP catabolites, namely the oxypurines (hypoxanthine, xanthine, and uric acid), because of an impairment between ATP production and consumption consequent to mitochondrial malfunctioning and activating the purine nucleotide catabolic pathway [12,13,14,15]. Concomitant to the energy crisis, an increase in oxypurine formation has also been shown, as has an increased formation of the stable end-products of nitric oxide metabolism (nitrite and nitrate), suggesting a close link among mitochondrial malfunctioning, energy metabolism impairment, and the insurgence of nitrosative stress [16,17]. Hence, measuring these compounds in cell culture media may be indirectly indicative of the correct cell energy balance and metabolism.

In previous studies, we have shown that the aforementioned compounds have significantly higher levels in the follicular fluid of infertile females compared to those measured in the follicular fluid of control fertile subjects, suggesting that poor oocyte biochemical quality deeply affects reproductive capacity [18]. To date, there are no available data showing whether these compounds can be detected in human embryo culture media and, even more important, whether the potential changes in their concentrations have any type of relationship with the human embryo’s quality or clinical pregnancy. By measuring purines, pyrimidines, nitrite, and nitrate in human embryo culture media, subsequently categorized into different morphological quality classes according to standard procedures, and further divided into those with successful or unsuccessful pregnancy, we here describe the potential relevance of the analyses of these metabolites for the selection of best human embryos to improve rates of successful ART.

## 2. Results

### 2.1. Metabolites in the Culture Media Correlate with Morphological Human Embryo Quality

The analysis of the embryo culture media, independently of the human embryo quality, revealed that hypoxanthine, uric acid, nitrite, and nitrate were detectable in most of the samples assayed, whilst xanthine, nucleosides (inosine, guanosine, and adenosine), and pyrimidines were undetectable or below the corresponding detection limits.

The morphological assessment of embryos allowed their categorization, according to the Gardner grading system, into those of high grade (A + B, n = 54) and those of low grade (C + D, n = 12). The levels of catabolites released during embryonic development that could be measured in the human embryo culture media, are illustrated in Figure 1 and Figure 2, where embryos were divided into those of high or low morphological quality according to the posterior classification using the Gardner scale.

Panels (A–C) of Figure 1 indicate that the concentrations of the two detectable oxypurines (hypoxanthine and uric acid) produced during embryonic development, as well as the sum of these two compounds derived from purine nucleotide catabolism (mainly ATP), were significantly higher in the culture media of human embryos of low morphological grade (C + D), suggesting a potential imbalance between ATP production and consumption or an accelerated rate of metabolism. It is worth underlining that, as it is clearly visible from Figure 1, the condition of “no detectability” for hypoxanthine and uric acid (occurring in 27/54 samples) was only found in high-grade embryos. Therefore, it is highly reasonable to affirm that the undetectability of these compounds in human embryo culture media is associated with high-quality embryo morphology.

Panels (A–C) of Figure 2 report the concentrations of nitrite (A), nitrate (B), and nitrite + nitrate (C) in the culture media of human embryos of high or low morphological grade. Both nitrite and nitrate were quantifiable in the media of the vast majority of both high- and low-quality embryos, with those of low grade having significantly higher values of these stable end-products of nitric oxide metabolism at 5 days post-fertilization (q < 0.001).

### 2.2. Metabolite Levels in the Culture Media Correlate with Successful Pregnancy and Live Births

To evaluate whether the concentrations of hypoxanthine, uric acid, the sum of oxypurines, nitrite, nitrate, and nitrite + nitrate may be predictive of successful pregnancy, we clustered culture media of human embryos according to the occurrence of clinical pregnancy, independently from the morphologically assigned embryo quality. Preliminarily, it is worth underlining that, in the high-quality group, 37/54 blastocysts were implanted, with pregnancy occurring in 15/37 implanted blastocysts (40.5%). In the low-quality group, 5/12 blastocysts were implanted, with pregnancy occurring in 3/5 implanted blastocysts (60%). Therefore, based on the embryo quality, a total of 42 human embryos (37 A + B-graded; 5 C + D-graded) were implanted. Of these, 17 were successful (40.4%), and 25 had unsuccessful pregnancies (59.5%, *p* < 0.001). These two new groups of human embryos were further utilized to categorize the results of the catabolites in the embryo culture media into those with successful or unsuccessful pregnancies. It is fundamental to underline that all successful pregnancies ended with live births and healthy newborns.

Figure 3 shows the concentrations of hypoxanthine (A), uric acid (B), and the sum of oxypurines (C) accumulated in the media of fertilized human embryos (during 5 days of culture) with subsequent successful or unsuccessful pregnancy. Higher releases of hypoxanthine (q < 0.02), uric acid (q < 0.01), and the sum of oxypurines (q < 0.001) were measured in the culture media of embryos with pregnancy failure, suggesting that this group of embryos is characterized either by cell energy imbalance or by an accelerated metabolic rate, in both cases critical to obtain a successful pregnancy. The condition of “no detectability” for hypoxanthine and uric acid occurred in 8/17 culture medium samples of embryos with successful pregnancies and only in 2/25 in those with unsuccessful pregnancies. Therefore, it is highly reasonable to conclude that the undetectability of these compounds in human embryo culture media is associated to successful pregnancy.

As shown in Figure 4, panels (A–C), higher concentrations of the stable end-products of nitric oxide metabolites (nitrite and nitrate) were released into the culture media of implanted human embryos with unsuccessful pregnancy, suggesting the possibility of nitrosative stress occurrence during human embryo development, that negatively influences rate of successful pregnancy and subsequent live births.

### 2.3. Release of Catabolites in the Culture Media May Be Useful to Biochemically Grade Embryo Quality and to Select the Best Embryos for Implantation

To evaluate whether release (hypoxanthine, uric acid, sum of oxypurines, nitrite, nitrate, and nitrite + nitrate) of specific compounds in human embryo culture media during human embryo development may have a potential utility to biochemically grade embryos and/or select embryos with the best probabilities of successful pregnancy, we calculated the Receiver Operating Characteristic (ROC) curves, either when metabolic data where categorized in the two groups of high and low embryo quality according to the morphological evaluation and the Gardner scale system, or when results were divided into two groups of successful and unsuccessful pregnancy on the basis of post-implantation embryo survival.

Figure 5 and Figure 6 show the ROC curves, respectively, for hypoxanthine (Figure 5A), uric acid (Figure 5B), sum oxypurines (Figure 5C), nitrite (Figure 6A), nitrate (Figure 6B), and nitrite + nitrate (Figure 6C) in two groups of morphologically high- and lowquality human embryos, clearly evidencing that these indexes of cell metabolism allow to biochemically cluster embryos into two distinct populations with different metabolic performances connected to their morphological characteristics. The statistical significances of the different AUCs indicate both high sensitivity and specificity, thereby corroborating the initial classification of human blastocysts into two groups of high and low quality on the basis of their morphological characteristics.

Since one of the goals of this study was to possibly add new criteria for blastocyst selection for subsequent in utero implantation, we calculated the ROC curves of the compounds detected in human embryo culture media when considering the two groups of human embryos with successful or unsuccessful pregnancies. As shown in Figure 7, only the sum of oxypurines and nitrite + nitrate had reasonably significant AUC values, indicating good sensitivity and specificity in the categorization of human embryos into those with successful or unsuccessful pregnancy.

## 3. Discussion

Following IVF, the analysis of purines, pyrimidines, nitrite, and nitrate (as stable end-products of nitric oxide metabolisms) in culture media during human embryonic development allowed a preliminary determination that only a minor number of the aforementioned, potentially measurable compounds were detectable in the majority of the samples assayed, independently of the morphological grade of the human blastocyst. That is, minimal amounts of the compounds detectable (hypoxanthine, uric acid, nitrite, and nitrate) are physiologically released in the culture media during human embryo development, opening the perspective that potential changes in their extracellular concentration may occur in the case of various conditions of altered cell metabolism. 

Using this approach of selectively measuring the aforementioned compounds, the posterior classification of the analyzed culture medium samples into two groups of blastocysts of high and low grade, according to morphological evaluation and classification using the Gardner grading system, clearly demonstrated that the altered morphology of human low-grade embryos is accompanied by significant changes in the release of ATP catabolites (hypoxanthine, uric acid, and sum oxypurines) and products of nitric oxide metabolism (nitrite and nitrate). This suggests that the levels of catabolites released by the two groups of high and low morphological quality embryos may be due to faster metabolic rates in the group of human low-grade embryos, potentially suggesting that accelerated metabolism may be causative of the increase in cell oxypurine release [19,20]. However, it should be taken into consideration the possibility that this phenomenon is due to incorrect mitochondrial functioning in low grade embryos, rendering the amount of mitochondrially produced ATP unable to satisfy cell energy demand, thus causing the activation of the purine nucleotide catabolic pathway and leading to a higher release of oxypurines [21,22]. In this light, the higher nitrite and nitrate release may further support the hypothesis of mitochondrial dysfunction characterizing the human low embryo quality group. Indeed, under various experimental conditions, overproduction of nitric oxide metabolites has been invariably associated with incorrect functioning of mitochondria [23,24,25], even though minimally adequate levels of these compounds, particularly nitric oxide and nitrite, are fundamental as signaling molecules to regulate crucial processes of cell metabolism [26,27,28].

The study also discovered that when the aforementioned compounds were assigned to implanted human embryos (of which 37 were rated A + B and 5 were rated C + D) with successful and unsuccessful pregnancy, significantly higher concentrations of ATP catabolites and nitric oxide metabolism were found in the culture media of the latter group. Even more important is that implanted human embryos developing into successful clinical pregnancies were all associated with live births and healthy newborns. This result clearly implicates that the biochemical evaluation of compounds connected to energy and nitric oxide metabolisms adds potentially invaluable information for the selection of those human embryos with the highest possibilities not only to develop into successful pregnancies but also to end with livebirths generating healthy newborns. 

It is also worth noting that, according to the simple morphological examination, 37/42 implanted human embryos were classified in the highest A + B rank of the Gardner grading system, implying that they all have the same chances of a subsequent successful pregnancy. The analysis of oxypurines and nitric oxide metabolites released into the culture media for 5 days following IVF by these 37 A + B human embryos, revealed that those with unsuccessful pregnancy had significantly higher catabolite levels (q < 0.01, for any of the parameters considered). This result confirms previous observations indicating that embryos of the same morphological quality do not always share the same metabolic characteristics [29].

Previous studies, aiming to apply an indirect metabolic evaluation of the embryo through the determination of various compounds in their culture media, mainly focused on quantifying the consumption of energy substrates [30,31,32] and/or the consumption and/or release of amino acids [33,34,35]. In both cases, since most of these compounds are normal constituents of the embryo culture media, it is very often necessary to measure little changes in the concentration of the compounds of interest (in the order of micromoles or fractions of micromoles) over a basal value of hundreds/thousands of times higher order of magnitude (tenths to hundreds of millimoles, depending on the compound considered). To complicate the situation, it should be considered that, while glucose may only be consumed by embryos, amino acid concentrations in embryo culture media are the result of a balance between the amount consumed for anabolic reactions and that produced by catabolic reactions (such as protein turnover), thus increasing the variability of the results referring to amino acid evaluation [35]. 

In the present study, we took into consideration only compounds that are not (or should not be) present as normal components of the culture media for growing human embryos. In fact, none of the compounds in the purines and pyrimidines classes we planned to measure are usual components of embryo growth media; that is, their measurements are performed over a basal value equal to zero. This renders the result of the analyses more trustworthy with a decrease in the dispersion of the data and with the certainty that, if measurable at the end of the period of in vitro embryo development, any quantity found is exclusively imputable to cell catabolic activity. Surprisingly, this situation did not apply to nitrite and nitrate. In fact, when we analyzed the different lots of growth media before their use for in vitro embryo development, we found variable amounts of both nitrite (ranging from 0.72 to 35.00 μmol/L) and nitrate (ranging from 9.51 to 76.61 μmol/L), derived either from spontaneous deamination and oxidation of amino acids or from a small pool of impurities possibly present in the batches of amino acids used for the growth media composition. Of course, in the data presented, the concentrations of either nitrite or nitrate measured in the culture media after 5 days of embryonic development were subtracted from the values measured in that specific growth medium lot at zero time.

Since the results of oxypurines may be interpreted as an indirect index of human blastocyst metabolic rate, with increased release of poorer embryos for both quality and successful pregnancy, our data seem to suggest that the higher the metabolic rate, the poorer the embryo quality and the probability of successful pregnancy. Data from the literature have not yet reached a consensus on this specific issue, so various studies are in line with our findings and others support the opposite indication [35,36,37]. On the other hand, our findings concerning the increased release of oxypurines in human embryos with a lower Gardner’s grade and failed pregnancy could be interpreted as an imbalance in ATP production and consumption, resulting in an increased rate of purine nucleotide catabolism and higher production and release of oxypurines (as end-products of this catabolic pathway). This condition has generally been found in the case of mitochondrial malfunctioning when mitochondrially produced ATP is insufficiently generated via oxidative phosphorylation because of decreased efficiency of the mitochondrial electron transport chain [38,39,40]. If this is the correct interpretation, then the mitochondria of human embryos with a worse Gardner’s grade and unsuccessful pregnancies did not properly work during the preimplantation period, leading to a higher release of oxypurines in the culture media.

## 4. Materials and Methods 

### 4.1. Patients’ Characteristics and Protocols for Ovarian Stimulation

The study was conducted according to the Declaration of Helsinki for medical research involving human subjects. All study participants provided informed written consent prior to study enrollment. Twenty-eight patients aged between 30 and 49, unable to obtain pregnancy after at least one year of unprotected sexual intercourses, with less than two previous ART cycles, were recruited and included in the study at the Alma Res Fertility Centre (Rome, Italy), from January to November 2021, with the approval of the Alma Res Ethical Committee (approval number AREC0319IVF). Women underwent a gonadotropin-releasing hormone (GnRh) antagonist stimulation protocol. Ovarian stimulation began on the second day of the menstrual cycle with recombinant FSH injections (Gonal-F by Merck Australia) or urinary hMG (Meropur by Ferring Switzerland). Transvaginal ultrasound and serum estradiol levels were used to monitor cycles, and a GnRh antagonist (Cetrotide by Merck Australia) was administered when follicles reached a size of ≥13 mm to prevent premature ovulation. When at least three follicles reached an approximate size of ≥18 mm each, they were aspirated 36 h after the injection of r-HCG. Transvaginal ultrasound-assisted follicle aspiration was performed with a vacuum pump.

### 4.2. Fertilization Procedures, Embryo Culture, Evaluation of Biochemical Pregnancy, Clinical Pregnancy, and Birth Rate

All patients underwent intracytoplasmic sperm injection (ICSI). Two hours following oocyte retrieval, oocytes were denuded from surrounding cumulus cells using a hyaluronidase solution (Vitrolife AB, Göteborg, Sweden) and inseminated via ICSI 2 h later. 

Injected oocytes were immediately transferred to individual culture wells with 50 µL Global Total Lp Medium (CooperSurgical, Inc., Trumbull, CT, USA) overlaid with 9 mL of Ovoil (Vitrolife AB, Göteborg, Sweden) in a pre-equilibrated GPS embryo dish (CooperSurgical, Inc., Trumbull, CT, USA). Embryos were cultured in a K-System incubator at 5% O_2_, 6% CO_2_, and 89% N_2_ at 37 °C, and the time of ICSI was set as the insemination time. 

On day 1, fertilizations were evaluated, and zygotes were transferred into a pre-equilibrated GPS embryo dish with fresh culture medium. The embryos remained in this plate until day 5 of development, when they were assessed for transfer or vitrification. On the same day, the number of developed blastocysts was counted, and their quality was evaluated in accordance with the Gardner blastocyst grading system [3]. Embryos grading higher than BB were considered to be of high quality (grade A and grade B embryos), while embryos grading lower than BB were categorized as being of bad quality (grade C and grade D embryos). The culture media of each embryo were collected and frozen at −20 °C and then processed for the analysis of metabolites, as subsequently indicated. Samples showing any visible trace of oil contamination were discarded. 

For those patients who decided on the embryo transfer, the number of clinical pregnancies and live births was also considered and then matched with the embryo quality. Of the 66 embryos cultured, 10 single blastocysts were transferred in a fresh cycle, and the remaining blastocysts were vitrified as part of a freeze-all cycle. The embryos with the best morphology were selected for transfer. Additionally, the embryos transferred in a thawed cycle were included in the pregnancy rate analysis. Biochemical pregnancy was recorded after an hCG blood test and clinical pregnancies were later confirmed at 8 weeks by ultrasound and observation of fetal heartbeat.

### 4.3. Biochemical Analysis of Embryo Culture Media

Frozen samples of embryo culture media (ranging from 25 to 45 μL) were diluted to 200 μL with HPLC-grade water and subsequently analyzed in blind. The high-performance liquid chromatographic analysis of purines, pyrimidines, nitrite, and nitrate was carried out on a 100 μL sample, using a well-established method set up in our laboratory and fully described in previous studies [18,41,42]. The simultaneous separation and quantification of the compounds of interest were carried out using a Surveyor HPLC system (Thermo Fisher Scientific, Rodano, Milan, Italy) equipped with a highly sensitive diode array detector (5 cm light-path flow cell) and connected to a Hypersil C-18, 250 × 4.6 mm, 5 μm particles, and 120 Å pore size column (Thermo Fisher Scientific, Rodano, Milan, Italy), protected by a 15 × 4.6 mm guard column. Using the ChromQuest^®^ software package provided by the HPLC manufacturer, the various compounds in culture media samples were quantified at 260 (purines and pyrimidines) or 206 nm wavelengths (nitrite and nitrate), by taking into account the dilutions of each sample.

### 4.4. Statistical Analysis

Statistical analysis was performed using the GraphPad Prism program, release 8.0 (GraphPad Software, San Diego, CA, USA). The Kolmogorov–Smirnov test was used to determine the normality of the distribution. Since data of the compounds under evaluation were not normally distributed, the Kruskal–Wallis non-parametric 1-way ANOVA, followed by the false discovery rate using the two-stage linear step-up procedure of Benjamini, Krieger, and Yekutieli for multiple comparisons as the post hoc test, was used when embryos were morphologically divided into high and low grade or those with successful and unsuccessful pregnancy. A q-value of less than 0.05 was considered statistically significant.

Additionally, the Receiver Operating Characteristic (ROC) curves to evaluate the sensitivity and specificity of the different compounds of interest and to cluster embryo culture media samples into high or low morphological grades and successful or unsuccessful pregnancies were also calculated.

## 5. Conclusions

Results of this study suggest that the analysis of compounds derived from ATP catabolism (oxypurines) and nitric oxide metabolism (nitrite and nitrate) may be a useful tool to evaluate, biochemically, human embryo quality to increase the rate of successful pregnancies and livebirths (generating healthy newborns) in patients undergoing ART. These analyses allow to obtain information otherwise impossible to obtain with the morphological evaluation and are, therefore, additive to this classic embryonic classification. 

Notwithstanding these promising indications, there are undoubtedly some points of weakness concerning the solidity of our findings and the potentiality of application of the analyses in the laboratory setting of assisted reproductive centers. The first is the relatively low number of human embryos analyzed (particularly those with low scores in Gardner’s grading system), certainly requiring confirmation in a further study with a larger sample size. The second concerns the timing of these biochemical analyses, which are certainly longer than the morphological evaluation. The HPLC determination of oxypurines, nitrite and nitrate requires 30 min, with an additional 30 min before a new sample is analyzed. Therefore, only 12 samples/instrument/working day may be analyzed.

In conclusion, the biochemical/metabolic evaluation of human embryo culture media appears to be a highly promising tool to increase the rate of successful pregnancy, which is a great advantage for patients needing ART to overcome infertility problems.

## Figures and Tables

**Figure 1 ijms-24-00890-f001:**
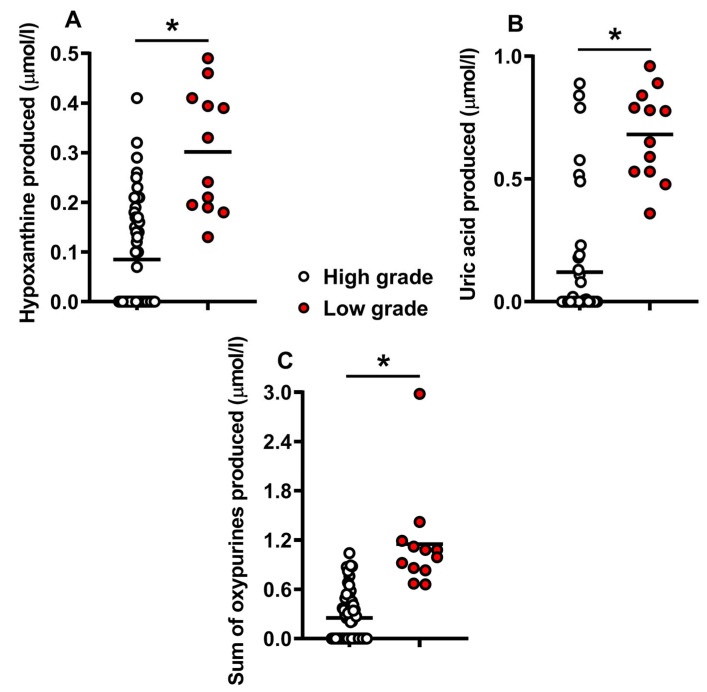
Concentrations of hypoxanthine (**A**), uric acid (**B**), and the sum of oxypurines (**C**) detected in culture media of human embryos at 5 days post-fertilization. The two groups of high-grade (n = 54, empty circles) and low-grade (n = 12, red filled circles) embryos were morphologically classified according to the Gardner scale. The mean values are represented by the horizontal bars. Sum oxypurines = hypoxanthine + uric acid. * Significantly different from high grade, q < 0.0001 (**A**–**C**).

**Figure 2 ijms-24-00890-f002:**
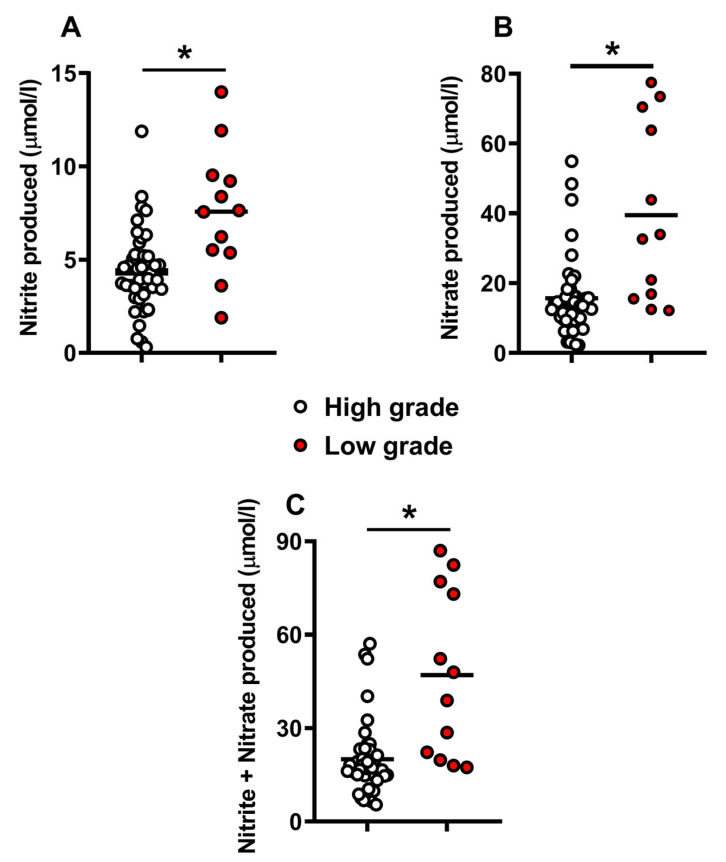
Concentrations of nitrite (**A**), nitrate (**B**), and nitrite + nitrate (**C**), as stable end-products of nitric oxide metabolism, detected in culture media of human embryos at 5 days post-fertilization. The two groups of high-grade (n = 54, empty circles) and low-grade (n = 12, red filled circles) embryos were morphologically classified according to the Gardner scale. The mean values are represented by the horizontal bars. * Significantly different from high grade, q < 0.001, (**A**–**C**).

**Figure 3 ijms-24-00890-f003:**
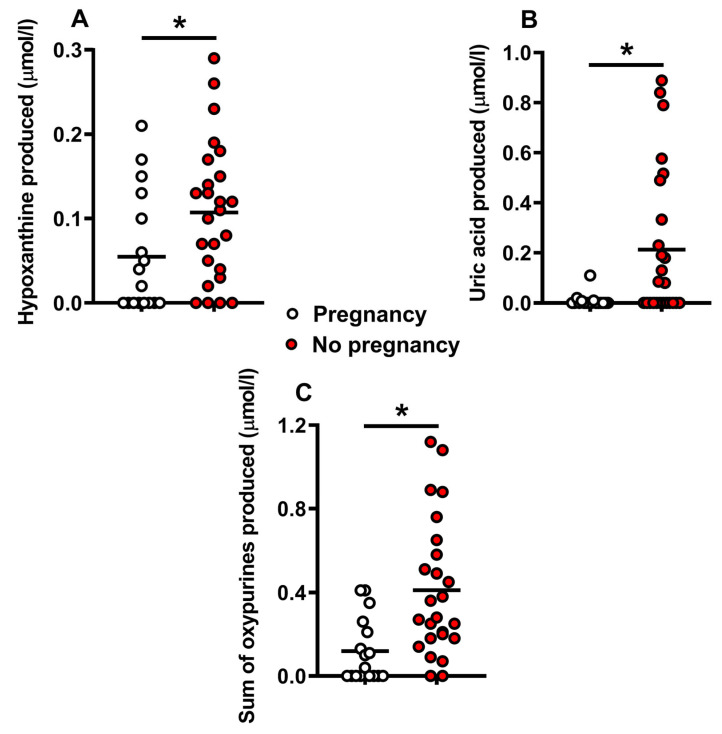
Concentrations of hypoxanthine (**A**), uric acid (**B**), and the sum of oxypurines (**C**) detected in culture media of human embryos at 5 days post-fertilization. Embryos were divided into those with successful (n = 17, empty circles) or unsuccessful pregnancy (n = 25, red filled circles), independently on the embryo grade. The mean values are represented by the horizontal bars. Sum oxypurines = hypoxanthine + uric acid. * Significantly different from high grade, q < 0.02 (**A**), q < 0.01 (**B**), and q < 0.0005 (**C**).

**Figure 4 ijms-24-00890-f004:**
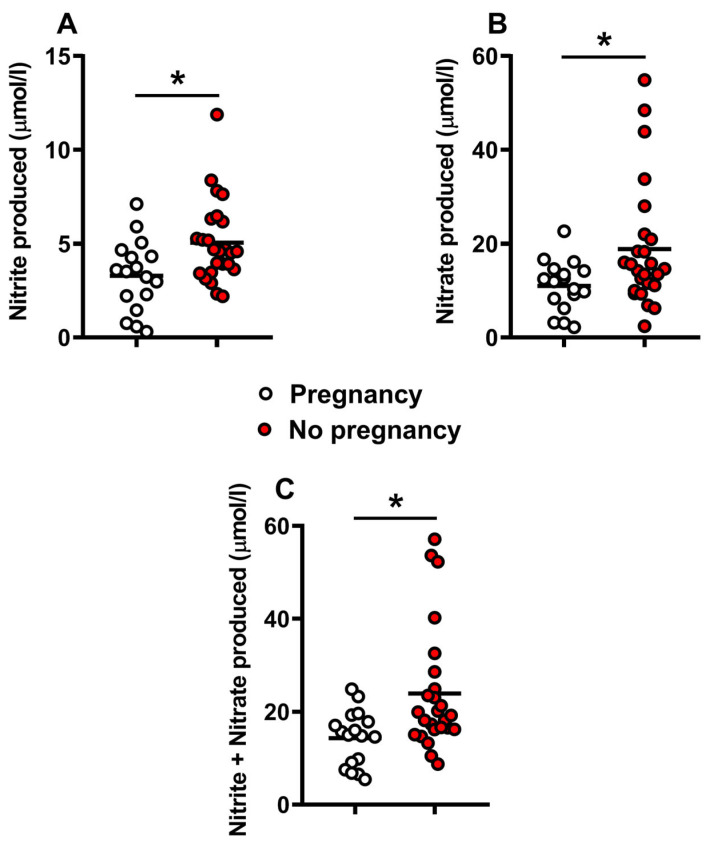
Concentrations of nitrite (**A**), nitrate (**B**), and nitrite + nitrate (**C**), as stable end-products of nitric oxide metabolism, detected in culture media of human embryos at 5 days post-fertilization. Embryos were divided into those with successful (n = 17, empty circles) or unsuccessful pregnancy (n = 25, red filled circles), independently on the embryo grade. The mean values are represented by the horizontal bars. * Significantly different from high grade, q < 0.01 (**A**), q < 0.05 (**B**), and q < 0.003 (**C**).

**Figure 5 ijms-24-00890-f005:**
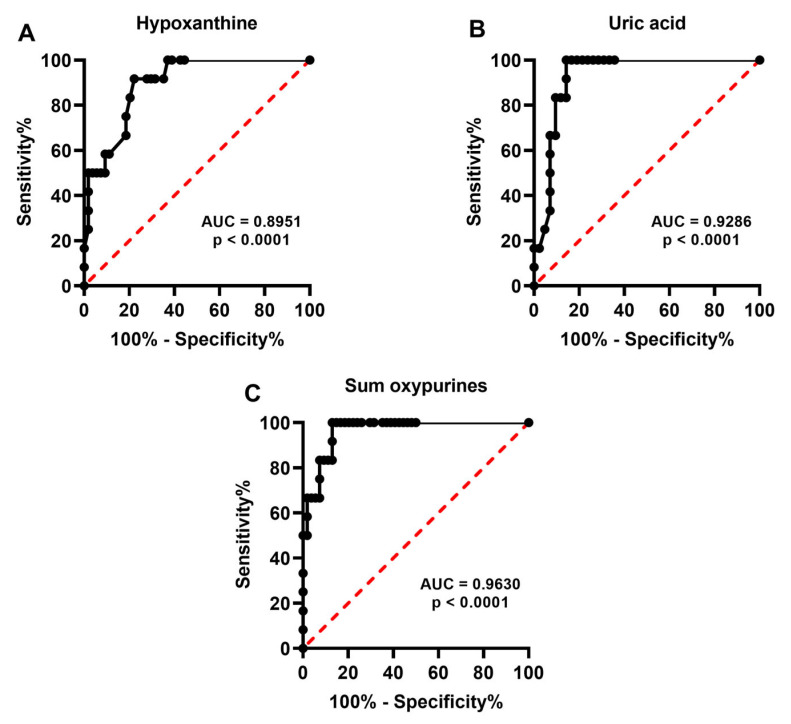
Receiver Operating Characteristic (ROC) curves of hypoxanthine (**A**), uric acid (**B**), and sum oxypurines (**C**) determined in the culture media of the two groups of human embryos categorized according to morphological quality using the Gardner scale system. The significance of the Area Under the Curve (AUC) is indicated in each panel. The sum of oxypurines = hypoxanthine + uric acid.

**Figure 6 ijms-24-00890-f006:**
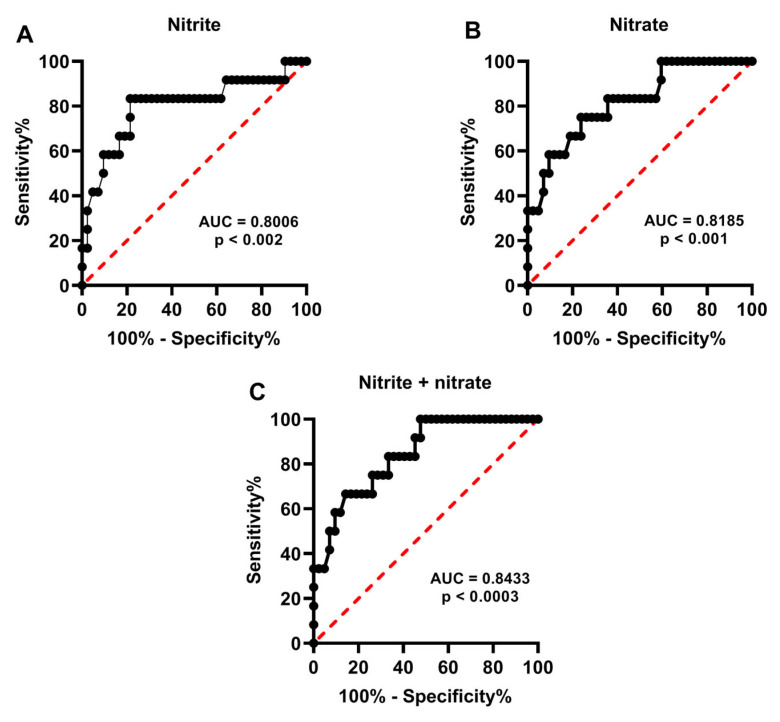
Receiver Operating Characteristic (ROC) curves of nitrite (**A**), nitrate (**B**), and nitrite + nitrate (**C**) determined in the culture media of the two groups of human embryos categorized according to morphological quality using the Gardner scale system. The significance of the Area Under the Curve (AUC) is indicated in each panel.

**Figure 7 ijms-24-00890-f007:**
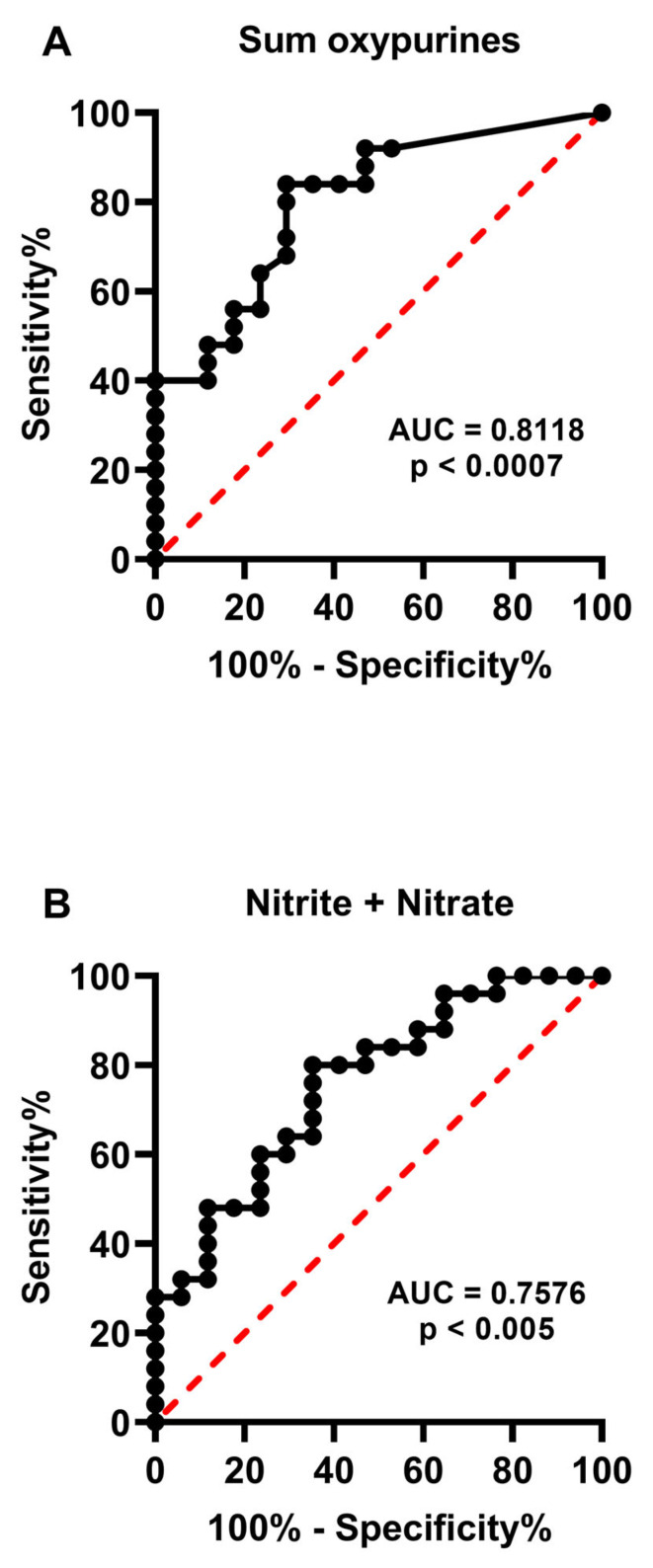
Receiver Operating Characteristic (ROC) curves of the sum of oxypurines (**A**) and nitrite + nitrate (**B**) determined in culture media of two groups of human embryos categorized according to morphological quality using the Gardner scale system. The significance of the Area Under the Curve (AUC) is indicated in each panel. The sum of oxypurines = hypoxanthine + uric acid.

## Data Availability

Data is unavailable due to privacy or ethical restrictions.

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
