# Peer review of "Metabolic Signature of Energy Metabolism Alterations and Excess Nitric Oxide Production in Culture Media Correlate with Low Human Embryo Quality and Unsuccessful Pregnancy"

_ijms, 2023, doi:10.3390/ijms24010890_

Round 1

Reviewer 1 Report

In this article, the authors investigated possible markers of predictive ability that give rise to a successful pregnancy in assisted reproductive techniques. In particular, they investigated some biochemical compounds in embryo culture media in relation to morphological assessments to increase the overall rate of ART.

The manuscript is interesting, clear and generally well written.  To my opinion, it can be accepted in the present form.

Author Response

Reviewer 1

We thank the reviewer the time spent in reading the manuscript and for her/his favorable comments on our data.

Reviewer 2 Report

In general the manuscript seems to be interesting, however, I have some important comments which should be considered by the authors:

1. While reading the article, I did not know which species it is based on until the method description. In my opinion "human" should appear in the title and in the introduction.

2. line 116 - in how many samples the products were not detected? Was it correlated with the quality of the embryo or with the volume of the medium sample?

3. The way the statistical differences on the graphs was shown is wrong. If two groups differ significantly, it is better to draw a bar above those groups and then put asterix above this bar. It should be applied to all the graphs.

4. "Blastogenesis" is an inappropriate word for the embryonic development up to the blastocyst stage.

5. Line 193 - what do you mean by "biochemically grade embryo quality"? I don't understand the title of this section.

6. Did you perform time-lapse monitoring of embryo development during this experiment? It would be great to show whether some development features correlate with the metabolimic data, especially that you suggest that metabolomic data probably come from faster metabolism of the embryo.

7. Line 293-294 - if you are not sure whether the media contain tested compounds, all the media before the culture should be also tested as zero samples.

8. Line 388 - it is impermissible that you collect and submit to the HPLC variable volume od the sample. If you sumbit higher volume - you have more compound! It is also easier to cross the minimal threshold.

Round 2

Reviewer 2 Report

The authors made the corrections to improve the manuscript. My only suggestion is to the point 8 of cover letter - I suggest to include to the methods description the information that when calculating the concentration of the compounds, the starting dillution of the sample was taken into account.

The manuscript is in the present form accepted for publication.